# Fast Tacrolimus Metabolism Does Not Promote Post-Transplant Diabetes Mellitus after Kidney Transplantation

**DOI:** 10.3390/ijms23169131

**Published:** 2022-08-15

**Authors:** Ulrich Jehn, Nathalie Wiedmer, Göran Ramin Boeckel, Hermann Pavenstädt, Gerold Thölking, Stefan Reuter

**Affiliations:** 1Department of Medicine D, Division of General Internal Medicine, Nephrology and Rheumatology, University Hospital of Muenster, 48149 Muenster, Germany; 2Department of Internal Medicine and Nephrology, University Hospital of Münster Marienhospital Steinfurt, 48565 Steinfurt, Germany

**Keywords:** kidney transplantation, immunosuppression, tacrolimus, NODAT, PTDM, diabetes, C/D ratio, metabolism

## Abstract

Post-transplant diabetes mellitus (PTDM) after kidney transplantation induced by tacrolimus is an important issue. Fast tacrolimus metabolism, which can be estimated by concentration-to-dose (C/D) ratio, is associated with increased nephrotoxicity and unfavorable outcomes after kidney transplantation. Herein, we elucidate whether fast tacrolimus metabolism also increases the risk for PTDM. Data from 596 non-diabetic patients treated with tacrolimus-based immunosuppression at the time of kidney transplantation between 2007 and 2015 were retrospectively analyzed. The median follow-up time after kidney transplantation was 4.7 years (IQR 4.2 years). Our analysis was complemented by experimental modeling of fast and slow tacrolimus metabolism kinetics in cultured insulin-producing pancreatic cells (INS-1 cells). During the follow-up period, 117 (19.6%) patients developed PTDM. Of all patients, 210 (35.2%) were classified as fast metabolizers (C/D ratio < 1.05 ng/mL × 1/mg). Fast tacrolimus metabolizers did not have a higher incidence of PTDM than slow tacrolimus metabolizers (*p* = 0.496). Consistent with this, insulin secretion and the viability of tacrolimus-treated INS-1 cells exposed to 12 h of tacrolimus concentrations analogous to the serum profiles of fast or slow tacrolimus metabolizers or to continuous exposure did not differ (*p* = 0.286). In conclusion, fast tacrolimus metabolism is not associated with increased incidence of PTDM after kidney transplantation, either in vitro or in vivo. A short period of incubation of INS-1 cells with tacrolimus using different concentration profiles led to comparable effects on cell viability and insulin secretion in vitro. Consistent with this, in our patient, collective fast Tac metabolizers did not show a higher PTDM incidence compared to slow metabolizers.

## 1. Introduction

To date, the calcineurin-inhibitor (CNI) tacrolimus (Tac) is the mainstay of immunosuppressive regimes after kidney transplantation (KTx), which usually consist of three components, namely a CNI, mycophenolate, and corticosteroids [1,2,3].

Unfortunately, Tac has a narrow therapeutic range. Due to the high intra- and inter-individual pharmacokinetics and pharmacodynamics of Tac, correct dose adjustment can be difficult [3]. Dependent on the concentration-to-dose (C/D) ratio, a slow (C/D ratio ≥ 1.05 ng/mL × 1/mg) or fast (C/D ratio < 1.05 ng/mL × 1/mg) Tac metabolism of treated individuals can be differentiated. This could be important because it has been shown by us and others that fast Tac metabolism is significantly associated with unfavorable renal allograft outcomes [4,5,6]. Although the exact mechanisms are still being researched, it is clear that Tac has direct toxic effects on pancreatic ß-cells and is therefore diabetogenic [4]. Nevertheless, these toxic effects are linked to pre-existing ß-cell-dysfunction, which might be induced by glucolipotoxicity [7]. Tac might act as an accelerator of pre-existing β-cell damage [8]. In rats, high-dose Tac treatment (10 mg/kg/day) decreased insulin mRNA transcription and reduced insulin production due to high levels of FK506-binding protein-12 in pancreatic β-cells [9]. Tong et al. found that high Tac concentrations inhibited insulin release and induced apoptosis in ß-cells [10]. Furthermore, CNI-induced hypomagnesemia is associated with post-transplant diabetes mellitus (PTDM), although the underlying mechanisms are still unknown [11]. Tac is therefore one important driver for PTDM. PTDM is frequently observed after KTx and is also associated with inferior outcomes such as infections and cardiovascular events [8]. The cellular mechanisms that lead to PTDM resemble those leading to type 2 diabetes mellitus to a considerable extent [12]. Risk factors for PTDM include obesity, dyslipidemia and insulin resistance [8].

We hypothesized that fast Tac metabolizers are at higher risk for PTDM due to the higher CNI toxicity of pancreatic ß-cells, since this is also true for renal vascular endothelial and tubular cells [13]. To investigate the effects of fast Tac metabolism on the development of PTDM, we performed a retrospective data analysis of 679 patients on the one hand. On the other hand, we modeled different tacrolimus exposure kinetics in slow and fast metabolizers via cell culture experiments in insulin-secreting pancreatic cells (INS-1 rat pancreatic insulinoma cell line).

## 2. Results

### 2.1. Retrospective Analysis of Patient Cohort

Of the 679 patients, 83 (12.2%) already had diabetes prior to KTx and were therefore excluded.

The remaining 596 patients had a median age of 51.7 years (IQR 21.2), and 351 (58.9%) of the patients were male. The mean BMI was 25.3 ± 4.1 kg/m^2^ (Table 1).

Of the 596 patients without diabetes at the time of KTx, 117 (19.6%) developed PTDM within a median onset time of 4.0 months after KTx (IQR 16.4 months). Of these, 37 (31.6%) showed lasting insulin dependency. A total of 210 (35.2%) of the 596 were classified as fast Tac metabolizers. Daily Tac dose (6.0 mg (IQR 4.0) vs. 5.0 mg (IQR 4.9), *p* = 0.324), trough levels (7.6 (3.2) ng/mL vs. 7.8 (3.5) ng/mL, *p* = 0.324) and C/D ratio at three months (1.54 ± 0.97 ng/mL × 1/mg vs. 1.65 ± 1.06 ng/mL × 1/mg, *p* = 0.467) did not differ between non-diabetic patients and those with PTDM (Table 2).

Patients who developed PTDM after KTx were significantly older (59.6 (18.1) vs. 49.9 (20.8), *p* = 0.010) and more obese (BMI 26.4 ± 3.7 vs. 25.1 ± 4.2, *p* < 0.001) than non-diabetics (Table 3).

The incidence of PTDM was not statistically different between slow and fast metabolizers (Log-rank *p* = 0.496, Figure 1).

A binary logistic regression confirmed BMI (*p* = 0.015) and recipients’ age (*p* < 0.001) as significantly associated with PTDM. The C/D ratio, which indicates Tac metabolism type, was not significantly associated with PTDM (*p* = 0.859) (Table 3).

### 2.2. Experimental Results

To determine whether INS-1 cell insulin secretion is affected by the Tac concentration profile used, INS-1 cells were exposed to a continuous Tac concentration and concentration profiles representing slow and fast Tac metabolism over 12 h (see Materials and Methods section for details).

The insulin secretion of INS-1 cells exposed to Tac in different concentration profiles (fast vs. slow vs. continuous serum profile) was comparable in all three groups (Continuous vs. Fast *p* = 0.323, Continuous vs. Slow *p* = 1.000, Slow vs. Fast *p* = 0.279, Figure 2).

The MTT assay was performed to ensure that cell viability did not differ between the Tac groups. It showed the comparable viability/metabolic activity of treated INS-1 cells between the three groups (Figure 3). As a control, we used INS-1 cells exposed to standard medium without Tac. Parallel to the Tac-treated groups, we changed medium every hour.

## 3. Discussion

We found no evidence in patient and cell culture data that a fast Tac metabolism, as depicted by C/D ratio, is associated with increased rates of PTDM. Fitting to that, neither Tac trough levels nor daily Tac dose was statistically associated with the development of PTDM.

Based on the knowledge that the underlying mechanism of Tac-associated PTDM is supposed to directly affect insulin secretion in pancreatic β-cells rather than inducing insulin resistance as an underlying mechanism for Tac-associated PTDM [9,10], we modeled the effects of slow and fast Tac metabolism on insulin-producing pancreatic cells (INS-1 cells) in vitro. The kinetics of Tac concentration used in cell culture experiments was determined in a former study showing that fast Tac metabolism is associated with increased nephrotoxicity in vitro and in vivo, respectively, and affected allograft function and allograft survival [4,14]. Because CNI toxicity is strongly related to the administered Tac dose relative to the trough level (C/D ratio) and thus to higher peak Tac levels, a higher risk of Tac renal toxicity in fast metabolizers is understandable [15,16]. However, the toxic action of Tac is not limited to renal cells and function, but, amongst others, is also apparent in pancreatic β-cells [10,12]. One mechanism for the diabetogenic effect of Tac might be the inhibition of the calcineurin-nuclear factor of activated T-cells (NFAT) pathway. This inhibition leads to the dysregulation of the insulin gene and alters β-cell mass, which was demonstrated in mice [17]. In contrast to the CNI cyclosporine A, Tac augments diabetogenic effects in patients with metabolic syndrome and present glucolipotoxicity [18]. Considering that, we treated INS-1 cells for our experiments with glucose and palmitate before exposure to Tac according to Trinanes et al. [19].

Because insulin release and the viability of INS-1 cells treated with different Tac protocols did not differ, peak Tac concentration does not seem to be a driving force for PTDM, unlike in nephrotoxicity. Notably, trough levels, daily Tac dose and C/D ratio were not different between patients who developed PTDM and those who did not (Table 2).

Rather, the development of PTDM in the clinical setting is determined by older age and higher BMI, as is known from other studies in KTx patients [20] and is also reproducible in our binary logistic regression model (Table 3). All patients in this study were medicated with steroids as one component of the triple immunosuppressive regimen in a standard maintenance dosage of 5 mg per day after 100 mg prednisolone for 3 days initially and 20 mg per day within the first month after KTx. Nevertheless, possible steroid pulses as therapy for allograft rejection were not taken into account. Since it has been demonstrated that KTx recipients with fast Tac metabolism are at higher risk of allograft rejection [4], it can be assumed that fast metabolizers even tended to be exposed to higher steroid dosages than slow metabolizers.

This study entails several further limitations. First, it cannot completely be excluded that some of the patients with PTDM already had undiagnosed diabetes before KTx. Because we used routine clinical data, we do not have data on the Tac peak level of patients. However, we know from a former study that fast metabolizers develop higher peak concentrations [14]. Furthermore, a retrospective data analysis on PTDM can only be hypothesis-generating. Regarding the experimental part of this study, the experiments did not model long-term effects of Tac exposure on pancreatic cells.

## 4. Materials and Methods

### 4.1. Study Cohort

We retrospectively analyzed a total of 679 KTx adult patients receiving an immunosuppressive regimen containing immediate-release (IR) or extended-release (ER) Tac with respect to the C/D ratio and classified them into slow versus fast Tac metabolizers based on the 3-month C/D ratio, as investigated in former studies (slow: C/D ratio ≥ 1.05 ng/mL × 1/mg, fast: C/D ratio < 1.05 ng/mL × 1/mg) [4,21]. Six hundred and forty-seven (95.3%) of the patients received IR Tac and thirty-one (4.7%) ER Tac. As well as Tac, the immunosuppressive triple regimen contained mycophenolate and steroids. The induction therapy was chosen according to the immunological risks of the patients.

From this cohort, 117 (17.2%) patients developed PTDM within the follow-up period (median 4.7 years (IQR 4.2 years)); a further 83 (12.2%) patients already suffered from diabetes before transplantation and were therefore excluded from further analyses. Moreover, we excluded from the study recipients < 18 years of age, pregnant women and patients with uncontrolled infection or active malignancy. PTDM was defined as persistent hyperglycemia that did not resolve within the first year after kidney transplantation in patients who did not suffer from diabetes before transplantation [22]. PTDM was defined based on the criteria for defining diabetes mellitus type 2 in the general population (ADA, WHO).

Written informed consent was obtained from all patients to record their data at the time of KTx. Data was taken from patients’ clinical files and personal information was anonymized prior to the analysis. This study was performed in accordance with the Declaration of Helsinki and the International Conference on Harmonization Good Clinical Practice guidelines and approved by the local ethics committee (EthikKommission der Ärztekammer Westfalen-Lippe und der MedizinischenFakultät der WestfälischenWilhelms-Universität, 2014-381-f-N).

### 4.2. Cell Culture

INS-1 832-13 cells (Duke Molecular Physiology Institute, Duke University, Durham, NC, United States), which were isolated from a rat insulinoma induced by X-rays and can secrete insulin, were used as a model for pancreatic β-cells [23]. INS-1 cells were cultivated in RPMI-1640 medium (Sigma-Aldrich, Darmstadt, Germany) supplemented with 10% fetal calf serum (FCS), 1% antibiotics (Pen/Strep), L-Glutamine, sodium-pyruvate and 2-mercaptoethanol (all supplements purchased from Sigma-Aldrich, Darmstadt, Germany) and were cultured at 37 °C and 5% CO_2_.

INS-1 cells were grown in 12-well or 96-well plates until 80% confluence followed by treatment with Tac (Prograf^®^i.v., Astellas, Munich, Germany) diluted in medium or with medium only as a control over 12 h. Tac working solutions were freshly prepared by appropriate dilution of stock solution in the culture medium [18]. According to the work of Rodriguez-Rodriguez et al., who showed that the toxicity of TAC depends on the pre-existence of β-cell dysfunction, we modeled glucolipotoxicity to the INS cells with glucose (22 mmol/L) and palmitate (100 μmol/L) for 5 days before we started the 12 h Tac time trials [5,8].

For 12 h time trials with exposure to the different Tac kinetic profiles, we applied Krebs–Ringer buffer (KRB) medium (own completion).

The culture medium of 12-well plates was changed every hour using the indicated Tac concentrations between 6 and 15 µg/mL that were based on our previous study [14], as shown in Table 4.

The medium of each well was collected after each hour and stored at −20 °C. Insulin concentrations of the medium samples were measured by rat insulin ELISA assay (Crystal Chem, Zaandam, Netherlands, LOT# 19JLRI094) with an assay range of 0.156–10.0 ng/mL according to the manufacturer’s instructions. Samples above the concentration limit for the test were re-measured after 10-fold dilution in calibrator diluent RD6-10 reagent according to the manufacturer’s specifications.

After the incubation period of 12 h, the culture medium was removed from the 96-well plate, the cells were washed three times with PBS, and then the samples were prepared for MTT assay, as described below. All samples were tested in triplicate wells.

### 4.3. MTT Test

Cell viability was assessed by a colorimetric assay, which is based on the conversion of dissolved yellow 3-[4,5-dimethylthiazol-2-yl]-2,5-diphenyltetrazolium bromide (MTT) to insoluble purple formazan by cleavage of the tetrazolium ring by mitochondrial dehydrogenases of living cells, as described in detail in former studies [14,24]. After Tac treatment for 12 h, the medium was carefully removed and replaced by 200 µL of fresh complete cell culture medium. Ten mL of MTT solution was added to each well and the cells were incubated again for 3 h. The medium was then removed and 100 µL of lysis buffer was added to each well. The plates were shaken for 10 min to destroy the cell structure and dissolve the blue formazan dye. Finally, absorbance was measured using an automated microtiter plate reader (Infinite M200; Tecan, Männedorf, Switzerland) [14].

### 4.4. Statistical Analysis

Data were analyzed using IBM SPSS Statistics 27 (IBM Corp., Armonk, New York, NY, USA). The results are expressed as a median with interquartile range (IQR) or a mean with standard deviation (SD). Non-continuous parameters were analyzed by Fisher’s exact test and continuous parameters were analyzed by the Mann–Whitney U-test if appropriate. A *p*-value ≤ 0.05 was considered statistically noticeable.

Repeated-measures ANOVA was applied to compare the insulin concentration in the cell culture medium samples of INS-1 cells over twelve hours between Tac concentration kinetics of slow and fast Tac metabolizers and a continuous Tac concentration, respectively.

We analyzed the cumulative incidence of PTDM after KTx using Kaplan–Meier analysis and the Log-Rank test.

Binary logistic regression analysis was applied to identify potential independent predictors of PTDM development.

## 5. Conclusions

In conclusion, fast Tac metabolism, defined by a C/D ratio of <1.05 ng/mL × 1/mg, in contrast to kidney function, does not impair pancreatic β-cell function to a greater extent than Tac slow metabolism, neither in a patient collective of KTx patients nor in an experimental setting.

## Figures and Tables

**Figure 1 ijms-23-09131-f001:**
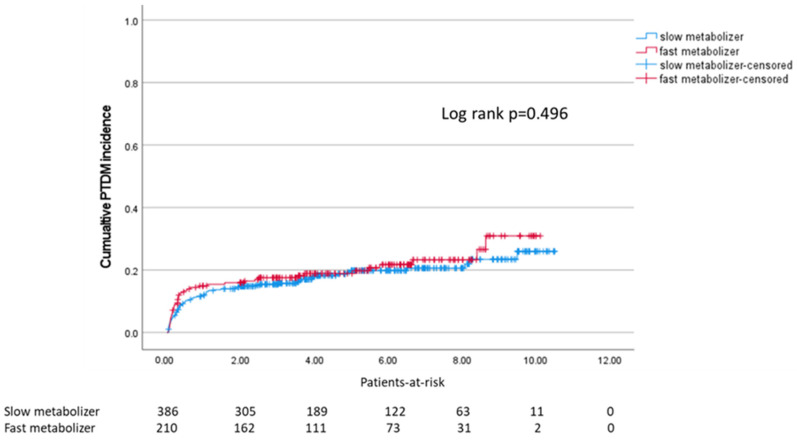
Kaplan–Meier survival plot for incidence of PTDM after transplantation in slow and fast metabolizers (median: 4.0 months).

**Figure 2 ijms-23-09131-f002:**
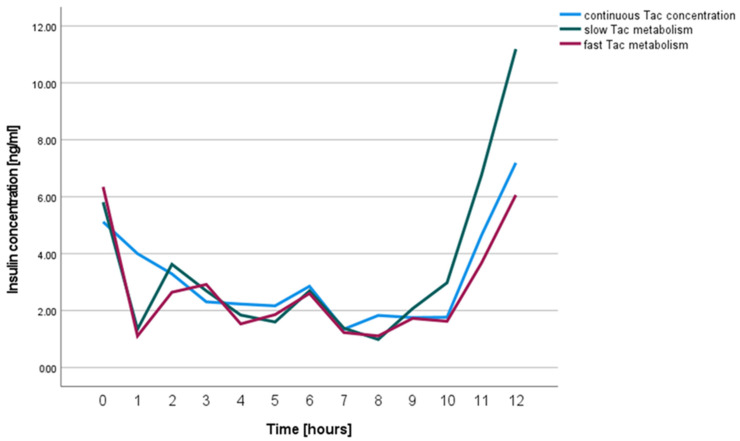
Insulin concentrations in cell culture medium supernatants of INS-1 cells treated with continuous Tac concentrations (blue), and variable concentrations modeling Tac kinetics of slow (green) and fast metabolizers (red).

**Figure 3 ijms-23-09131-f003:**
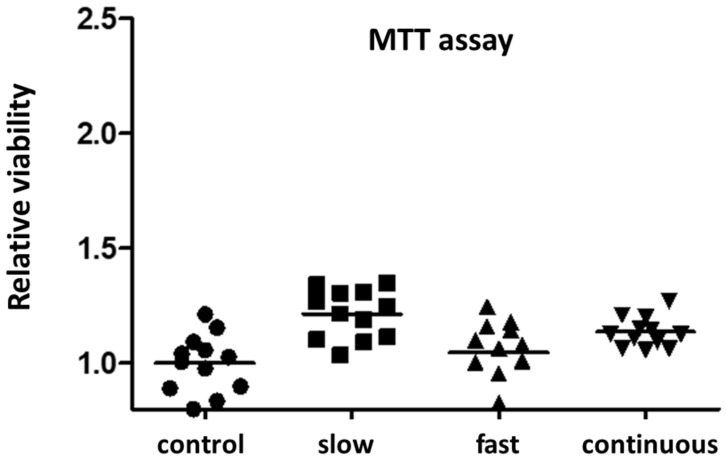
Relative viability of INS-1 cells treated with Tac concentrations using different incubation profiles related to a control of cultured INS-1 cells exposed to standard medium without Tac.

**Table 1 ijms-23-09131-t001:** Patients’ demographic and clinical characteristics at transplantation.

Variable	All	Non-Diabetic	PTDM	*p* Value
Patients n (%)	596 (100%)	479 (80.4%)	117 (19.6%)	-
Age at Tx * (years, median (IQR))	51.7 (21.2)	49.9 (20.8)	59.6 (18.1)	0.010 ^a^
Sex male n (%)	351 (58.9%)	284 (59.3%)	67 (57.3%)	0.258 ^b^
BMI kg/m^2^ (mean ± SD)	25.3 ± 4.1	25.1 ± 4.2	26.4 ± 3.7	<0.001 ^a^
Living donations n (%)	184 (30.9%)	167 (34.7%)	17 (14.5%)	<0.001 ^b^
Cold ischemic period (hours, mean ± SD)	8.4 ± 5.2	7.9 ± 5.3	9.3 ± 4.9	0.006 ^a^
Warm ischemic period (minutes, mean ± SD)	32.9 ± 8.4	32.8 ± 8.7	32.8 ± 8.2	0.705 ^a^
Dialysis vintage (months, mean ± SD)	62.1 ± 40.8	62.4 ± 41.4	60.7 ± 38.8	0.829 ^a^
Diagnosis of ESRD, n (%)				0.769 ^b^
- Hypertensive nephropathy	53	41	12	-
- Diabetic nephropathy	0 (0%)	0 (0%)	0 (0)	-
- Polycystic kidney disease	94	71	23	-
- Obstructive nephropathy	34	29	5	-
- Glomerulonephritis	215	181	34	-
- FSGS	19	15	4	-
- Interstitial nephritis	31	20	11	-
- Vasculitis	15	12	3	-
- Other	74	63	11	-
- Unknown	61	47	14	-

^a^ Mann–Whitney U-test. ^b^ Fisher’s exact test. Abbreviations: PTDM: Post-transplant diabetes mellitus; * Tx: transplantation; IQR: interquartile range; BMI: body mass index; SD: standard deviation; ESRD: end-stage renal disease; FSGS: focal segmental glomerulosclerosis.

**Table 2 ijms-23-09131-t002:** Patients’ outcome data.

Variable	All (n = 596)	Non-Diabetic (n = 479)	PTDM (n = 117)	*p* Value
C/D ratio after 3 months (mean ± SD)	1.56 ± 1.0	1.54 ± 0.97	1.65 ± 1.06	0.467 ^a^
Daily Tac dose (mg, median (IQR))	5.5 (4.0)	6.0 (4.0)	5.0 (4.9)	0.324 ^a^
Trough level (ng/mL, median (IQR))	7.7 (3.2)	7.6 (3.2)	7.8 (3.5)	0.324 ^a^
Fast metabolizers n (%)	210 (35.2%)	166	44	0.590 ^b^
Time to PTDM (years, median (IQR))	-	-	0.33 (1.37)	-
Insulin-dependent diabetes	-	-	37 (31.6%)	-

^a^ Mann–Whitney U-test. ^b^ Fisher’s exact test. Abbreviations: PTDM: post-transplant diabetes mellitus; C/D ratio: concentration-to-dose ratio; Tac: tacrolimus; IQR: interquartile range.

**Table 3 ijms-23-09131-t003:** Binary logistic regression for PTDM risk factor identification.

Variable	Regression Coefficient	Odds Ratio	95% CI	*p* Value
Recipient age	0.048	1.050	1.032–1.068	<0.001
BMI	0.064	1.066	1.013–1.122	0.015
C/D ratio	0.019	1.019	0.829–1.252	0.859

Abbreviations: CI: confidential interval; BMI: body mass index; C/D ratio: concentration-to-dose ratio.

**Table 4 ijms-23-09131-t004:** Based on the results of our former study, INS-1 cells were treated with the following Tac concentrations hourly for a period of twelve hours to model slow, fast and continuous Tac metabolism [17].

Time (Hour)	1.	2.	3.	4.	5.	6.	7.	8.	9.	10.	11.	12.
Slow Tac (µg/mL)	6	12	11	9	9	9	8.5	8.5	8	8	7	6
Fast Tac (µg/mL)	6	11	15	11	8	7	6	6	6	6	6	6
Continuous Tac (µg/mL)	8.5	8.5	8.5	8.5	8.5	8.5	8.5	8.5	8.5	8.5	8.5	8.5

## Data Availability

The datasets generated and/or analyzed during the current study are available from the corresponding author on reasonable request.

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
