# Peer review of "Fast Tacrolimus Metabolism Does Not Promote Post-Transplant Diabetes Mellitus after Kidney Transplantation"

_ijms, 2022, doi:10.3390/ijms23169131_

Round 1

Reviewer 1 Report

Dr. Jehn and colleagues report that fast tacrolimus (TAC) metabolism does not promote diabetes mellitus post kidney transplantation. However, the authors must address the following concerns regarding their experimental design and data reporting before the paper should be considered for publication in IJMS.

1.     The experimental modeling of fast and slow TAC metabolism in INS 1 cells is not an ideal model. What is the reason of using INS I cells rather than human islets?

2.     Insulin cells are constitutive for insulin release. This is different from type 2 diabetes. So measuring insulin release from a cell line that has an aberrant insulin release mechanism is totally faulty. Again it would be ideal to use islets from diabetic donors; at least rat islets from type 2 diabetes rats.

3.     In figure 2 authors show insulin concentration, which is secretion, not content. Their own reference showed that this treatment affects the content and does not affect secretion. Please explain why there is no TAC, no palmitate and no glucose group?

4.     Authors provide p value for figure 2, however they never mention that which time point p values? At 12 hr time point these look different. And also they discussed it poorly in the result sections. 

5.     In materials methods (4.2) authors refer another papers (4,5) which showed that the toxicity of TAC depends on the pre-existing beta cell dysfunction which is the reason they modeled glucolipotoxicity to the INS cell with glucose and palmitate. Are Ins I cells susceptible to this? Were they, in fact, damaged by this treatment? If not then this is not a toxicity model.

6.     Authors found that patients who developed PTDM after transplantation were significantly older and obese. Did they look at their fasting blood glucose levels before the transplant to determine whether they were pre-diabetic or not? This should be available in pre-transplant blood work on the day of surgery.

7.     In Figure 3 authors showed relative viability. However, these cells are transformed cells, so their viability is already enhanced. They should use islets.

Reviewer 2 Report

This paper deals with a possible role of tacrolimus metabolism kinetics and the development of post-transplant diabetes after kidney transplantation (PTDM). In order to investigate this topic, the authors analyzed retrospectively a cohort of 596 non-diabetic patients and performed some in vitro experiments using cultured beta-pancreatic cell line. In general, the authors try to transfer their own previous data on C/O ratio in nephrotoxicity to PTDM.

The experimental data are not sufficient to corroborate the hypothesis of an impact of Tac-metabolism o insulin secretion. Did they take into account cell division over 12 hours affecting the cell number? Since they report on apoptosis possibly confounding insulin secretion?

This manuscript contains a substantial number of inconstancies making it difficult to follow the author’s thoughts. Even if the data are clear and the conclusions are reasonable, the way of presentation needs profound revision. Beside this, the reviewer has the feeling that English editing is needed throughout the manuscript (for example see page 3, line 78).

My remarks in detail:

Abstract:

Page 1:

·         Line 19: 117 (17.2%): What does this number refer to? 117/596 (line 15) = 19.9 %

·         Line 24: The conclusion should be the last sentence of the abstract referring to bot, clinical and in vitro data.

·         Introduction: 

·         Line 32: Kidney transplantation rather than Ktx., please.

·         Line 32: A short reference to the substances mentions at the end of the sentence would be helpful (what is tacrolimus?).

·         Line 39: “shown by us and others“ requires multiple references, not only one to one’s own publications.

·         Line 44: β-cells or beta cells? Please make up your mind.

·         Line 50: “could be” rather than “is”

Page 2:

·         Line 50: Reference 9 deals with endothelial cells, not with pancreatic β-cells. Please stay consistent.

·         Line 54: What does the last sentence mean in the context of immunosuppression?

·         Line 58: The inconsistent enumeration of overall patients (here) and 596 non-diabetic patients (abstract and elsewhere) is confusing.

Page 3:

·         Table 2: Formatting must be revised: 7.6 (3.2)

·         Line 76: 117/596 = 19.6%. Please clarify.

·         Line 78: 37/596 = 6.2%. or 37/117 PTDM = 36,6% Please clarify.

·         Line 78: “of these” rather than “of them”? Please consult a native English speaker.

·         Line 78: “Two hundred and ten (35.2%) of the PTDM patients”. What does it mean? 210/117 = 179 %? Or 210 or all patients? Or “44 of PTDM-patients”?

Page 4:

·         Line 91: (BMI 25.1 ± 4.2 vs. 26.4 ± 3.7, p<0.001)). The other way around: the higher BMI first.

·         Figure 1: Please use more distinguishable colors.

·         2.2: Experimental results: This section needs much more detailed description: Aim of the experiment?

Page 5:

·         Figure 2: How do you explain the decrease during the first hour?

·         Figure 2 legend: Concentrations in what? Cell culture medium? Intracellular staining?

·         Figure 3: What does control mean? Untreated cells, serum alone,  positive control, negative control?

Page 6:

In general, the discussion is much to long for a relatively limited data set.

·         Line 126: “has been” rather than “was”

·         Line 129: The paper deals with PTDM and not with nephrotoxicity. Please strengthen.

·         Discussion of Akt an mTOR provokes the question why the researchers did not analyze this.

·         Line 144: Very general remarks, not topic of this study.

·         Line 158 - 1169: Age and BMI are confounders of this study. Does this paragraph really help to understand the mean message of the paper?

·          

Page 9

·         References 1 and 2: „Schutte-Nutgen“ oder „Schütte-Nütgen“?

Reviewer 3 Report

Well done study, showing that a TAC toxicity to the pancreas is a complex issue. Just a few minor additional data will be appreciated (if available):

- can C max (C2) of TAC concentration be presented and compared?

- did diabetic patients presented lower serum magnesium (Mg)  concentration? (it was suggested as additional PTDM promoting factor, relevant also for higher TAC exposure)

- was there are ary association with pre-transplant peritoneal dialysis vintage (expected as more frequent in older patients)?

Round 2

Reviewer 1 Report

Thanks for clarifying all the points.

Reviewer 2 Report

The authors have sufficiently addressed my suggestions.

Only one mirror point: Did you in corporate your comment 24 into the discussion?

Please do so.

Author Response

Thank you very much for your recent suggestions on our manuscript ijms-1829933 entitled  "Fast tacrolimus metabolism does not promote post-transplant diabetes mellitus after kidney transplantation", which had been submitted to the International Journal of Molecular Sciences and for giving us the opportunity to revise it again.

We have carefully revised our paper taking into account the comments of the reviewers and would like to respond to them in detail. We have implemented or discussed all the points requested. We also would like to thank the reviewers for their helpful and constructive comments to improve the manuscript.

Besides a clean version of the revised manuscript, we attached a version with all changes highlighted in yellow for clarity.

We are looking forward to your kind reply and would appreciate to find our paper to be published in the International Journal of Molecular Sciences.

Please note, that we added the co-author Dr. Göran Ramin Boeckel, who supported data collection which was necessary for the revision.

Sincerely,

Ulrich Jehn and Stefan Reuter, in behalf of the authors
